# Mechanism of Siponimod: Anti-Inflammatory and Neuroprotective Mode of Action

**DOI:** 10.3390/cells8010024

**Published:** 2019-01-07

**Authors:** Newshan Behrangi, Felix Fischbach, Markus Kipp

**Affiliations:** 1Department of Anatomy II, Ludwig-Maximilians-University of Munich, 80336 Munich, Germany; nbehrangi@gmail.com (N.B.); felix@famfischbach.de (F.F.); 2Department of Anatomy, University Medical Center, 39071 Rostock, Germany

**Keywords:** multiple sclerosis, neurodegeneration, inflammation, siponimod, progressive

## Abstract

Multiple sclerosis (MS) is a neuroinflammatory disorder of the central nervous system (CNS), and represents one of the main causes of disability in young adults. On the histopathological level, the disease is characterized by inflammatory demyelination and diffuse neurodegeneration. Although on the surface the development of new inflammatory CNS lesions in MS may appear consistent with a primary recruitment of peripheral immune cells, questions have been raised as to whether lymphocyte and/or monocyte invasion into the brain are really at the root of inflammatory lesion development. In this review article, we discuss a less appreciated inflammation-neurodegeneration interplay, that is: Neurodegeneration can trigger the formation of new, focal inflammatory lesions. We summarize old and recent findings suggesting that new inflammatory lesions develop at sites of focal or diffuse degenerative processes within the CNS. Such a concept is discussed in the context of the EXPAND trial, showing that siponimod exerts anti-inflammatory and neuroprotective activities in secondary progressive MS patients. The verification or rejection of such a concept is vital for the development of new therapeutic strategies for progressive MS.

## 1. Introduction

Multiple sclerosis (MS) can be clinically categorized into three groups: Relapsing-remitting, secondary progressive, and primary progressive. In most patients, the initial course of the disease is relapsing-remitting which is characterized by acute clinical attacks that are followed by complete or incomplete recovery, with a period of remission in between the attacks. Many patients with an initial relapsing-remitting disease course after several years develop secondary progressive MS which is characterized by a more or less continuous decline of neurological functioning, with or without occasional attacks. Primary progressive MS is characterized by the accumulation of clinical disability from the disease onset, without early relapses or remissions. 

MS is traditionally considered to be an inflammatory autoimmune disease of the central nervous system (CNS) which is mediated by an aberrant lymphocyte attack directed against CNS elements. Although the autoantigen has not yet been discovered, it is assumed that the autoreactive immune response in MS patients is directed against a distinct component (or various components) of the myelin sheath. Perivascular inflammatory infiltrates, oligoclonal immunoglobulin G in the cerebrospinal fluid, gadolinium-enhancing lesions on magnetic resonance scans, or the acute appearance of clinical symptoms are believed to be the direct consequence of focal inflammatory CNS lesions. Such inflammatory lesions can be found widespread within the white matter, however, several studies have clearly shown that diverse brain grey matter structures are equally affected [1,2]. The inflammatory attacks do not just destroy the myelin sheath (i.e., demyelination) but as well affect the integrity of neuronal structures such as axons, dendrites, synapses or even entire nerve cells [3,4]. 

Given its indisputable inflammatory character, neurodegeneration in MS is commonly considered to be a direct consequence of inflammatory attacks. Following this concept, recruited peripheral immune cells release inflammatory mediators leading to neuronal damage. However, some authors believe that inflammation and neurodegeneration are two separate aspects of MS, especially during the progressive disease stage. While several excellent review articles have been published addressing the assumed dichotomy of inflammation and neurodegeneration during progressive MS [5,6], in this brief article we aim to highlight another scenario of the inflammation-neurodegeneration interplay, that is: Neurodegeneration can trigger the formation of new, focal inflammatory lesions. Of note, such a proposed scenario has direct therapeutic consequences. If neurodegeneration indeed triggers inflammation, then neuroprotection would ameliorate both inflammation driven relapses and neurodegeneration driven disability progression. 

Such a proposed therapeutic principle is discussed in the context of the recently published EXPAND study demonstrating that the Sphingosine 1-phosphate receptor modulator, siponimod, does not just ameliorate the inflammatory aspect but also the degenerative aspect of secondary progressive MS. Of note, we do not propose that brain degenerative events trigger the development of encephalitogenic immune cells, but rather suggest that once encephalitogenic immune cells are present in the blood in relevant concentrations, CNS-intrinsic events might trigger their central recruitment and, thus, the development of focal inflammatory lesions. 

Before we give a brief introduction into the development of siponimod, we would like to point out that whenever we use the term “neurodegeneration” as a causative event during inflammatory lesion formation, we include “degeneration of the myelin sheath”.

## 2. Amelioration of Brain Cell Degeneration Might be Anti-Inflammatory

### 2.1. A Historical Perspective of Siponimod Development

Sphingosine 1-phosphate (S1P) is a bioactive sphingolipid that regulates a variety of physiological processes including lymphocyte recirculation and cardiac function. Most S1P effects are mediated via five G-protein-coupled S1P receptor subtypes referred to as S1P1–5 (originally termed EDG-1, 3, 5, 6, and 8) [7]. These receptors are differentially expressed on various cell types, including lymphocytes [8,9], cardiomyocytes [10,11] and brain cells (see Figure 1 for an overview of S1P1-5 expression in the CNS). In 2000, Kuppermann showed that S1P receptors regulate cell migration during vertebrate heart development [12], and two years later a similar pro-migratory effect of S1P receptors was demonstrated for CD4^+^ T cells [13]. Two years after, in 2004, Matloubian for the first time showed that the egress of lymphocytes from the thymus and the peripheral lymphoid organs is dependent on S1P1 [14]. Due to these observations, the anti-inflammatory potency of S1P-receptor modulators have been intensively investigated. 

The S1P-receptor modulator fingolimod, also called FTY720, induces a rapid and drastic deletion of T cells from the peripheral blood by inhibiting the egress of T cells from the thymus [15] and lymph nodes. By this mechanism, fingolimod prevents the entry of lymphocytes into the blood, and thus T cell infiltration into the CNS [16,17,18]. It has additionally been demonstrated that fingolimod can trigger lymphocyte apoptosis [16,19,20]. Consequently, preclinical studies show that fingolimod ameliorates pathology in several models of autoimmune diseases, including type 1 diabetes [21], adjuvant-induced arthritis [22], systemic lupus erythematosus [23] and, most importantly in the context of MS research, in different models of experimental autoimmune encephalomyelitis (EAE) [24,25]. In a number of clinical trials, it has been shown that fingolimod is well tolerated and associated with low relapse rates and lesion activity in relapsing-remitting MS patients [26,27,28,29]. Consequently, fingolimod was the first oral disease-modifying therapeutic agent to be approved for the treatment of MS. This pro-drug is rapidly converted in vivo into the active S-fingolimod-phosphate (FTY720-P) which is a potent agonist on S1P1, S1P3, S1P4 and S1P5 receptors. Since S1P-receptors are ubiquitinated and subsequently degraded when exposed to FTY720-P [30], the experimental and clinical efficacy of FTY720-P is thought to involve functional antagonism by persistent internalization and enhanced degradation of the S1P-receptor. Of note, its efficacy in MS and related animal models may in part be due to additional, direct effects within the brain. For example, a strong increase in S1P1 and S1P3 expression on reactive astrocytes was detected in active and chronic inactive MS lesions [31], whereas another study has suggested S1P5 expression in oligodendrocytes [32,33]. 

**Figure 1 cells-08-00024-f001:**
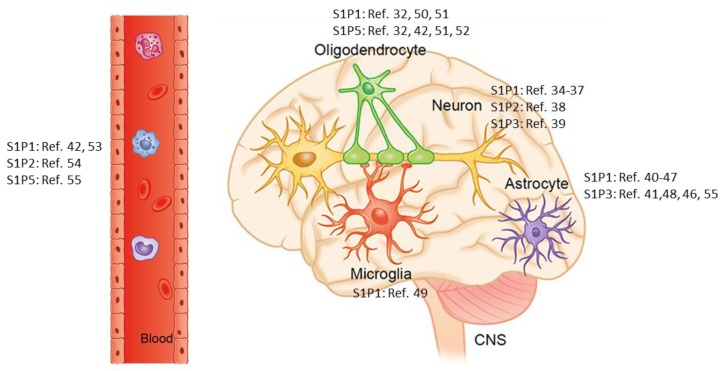
Sphingosine 1-phosphate receptor expression in the central nervous system: Schematic illustrating the expression of Sphingosine 1-phosphate (S1P) receptor subtypes 1–5 (referred to as S1P1–5) in the cells of the central nervous system. Note that S1P receptors are expressed by endothelial cells as well. References: [31,32,34,35,36,37,38,39,40,41,42,43,44,45,46,47,48,49,50,51,52,53,54,55]. The copyright license of this illustration is with the authors.

In general, fingolimod has a favorable benefit-risk profile [56]. However, a critical challenge of fingolimod therapy still remains in the initiation phases due to the risk of cardiac events. The first dose of fingolimod is associated with a decrease in heart rate and slowing of atrioventricular conduction [28,29,57]. The discovery of the S1P3 receptor mediating bradycardia in mice [58] prompted the search for S1P-receptor modulators devoid of S1P3 signaling. This effort led to the discovery of siponimod (also called BAF312), which is a selective modulator of S1P1 and S1P5 receptors. Siponimod was furthermore designed to have a relatively short elimination half-life that provides a rapid recovery of blood lymphocyte counts on stopping treatment, but would allow once-daily oral dosing [59].

### 2.2. Results of the EXPAND Study

In 2013, Selmaj reported the results of a phase 2 dose-finding study in patients with relapsing-remitting MS (RRMS). Siponimod reduced active brain lesion counts and the annualized relapse rate by approximately two-thirds, in a dose-dependent manner [60]. Due to the eminent medical need for having treatment options during progressive MS, a phase 3, randomized, parallel-group, double-blind, placebo-controlled, event-driven, and exposure-driven trial (EXploring the efficacy and safety of siponimod in PAtients with secoNDary progressive multiple sclerosis [EXPAND]) was conducted to investigate the efficacy and safety of siponimod in patients with secondary progressive MS (SPMS) [61]. Key inclusion criteria for subjects was being aged 18–60 years, having a history of RRMS following the 2010 revisions to the McDonald criteria [62], having a confirmed diagnosis of SPMS, having a moderate-to-advanced disability indicated by an Expanded Disability Status Scale (EDSS) score of 3–6 at screening, having documented EDSS progression in the 2 years before the study, and having no evidence of a relapse in the 3 months before randomization (see Figure 2). The primary endpoint of the EXPAND study was the time to 3-month confirmed disability progression, which was defined as a 1-point increase in EDSS if the baseline score was 3.0–5.0, or a 0.5-point increase if the baseline score was 5.5–6.5, confirmed at a scheduled visit at least 3 months later. From February 2013 to June 2015, 1105 patients were randomly assigned to the siponimod group, and 546 to the placebo group, respectively. As one main result of this study, there was a significant reduction in the 3-month confirmed disability progression, with 26% of patients in the siponimod group and 32% in the placebo group having a 3-month confirmed disability progression, which equaled a relative risk reduction of 21% compared with placebo. 

A number of secondary endpoints are relevant for this review article: First, the increase in T2 lesion volume from baseline was lower with siponimod than with placebo. Second, a higher number of patients receiving siponimod than placebo were free from gadolinium-enhancing lesions, and third, more patients receiving siponimod than placebo were free from new or enlarging T2 lesions. All these findings suggest a potent anti-inflammatory activity of siponimod in SPMS patients. Furthermore, brain volume decreased at a lower rate with siponimod (0.28%) than with placebo (0.46%). The only key secondary outcome that did not favor siponimod was time to 3-month confirmed worsening of at least 20% in the timed 25-foot walk (T25FW). In summary, siponimod attenuated the inflammatory activity in SPMS patients and at the same time ameliorated the degenerative aspect of the disease (i.e., disease progression). Notably, a subgroup analyses of the EXPAND study data suggested that the treatment effect of siponimod is most pronounced in patients with ongoing inflammatory activity. This finding is similar to what was seen with ocrelizumab in the ORATORIO trial in primary progressive MS, where greater benefit was detected in patients with gadolinium-enhancing lesions at baseline [63].

### 2.3. Possible Siponimod Mode of Action

As pointed out in the previous chapter, siponimod reduced the inflammatory activity (i.e., less gadolinium-enhancing lesions and new or enlarging T2 lesions) as well as the extent/progression of neurodegeneration (i.e., reduced disability progression and brain atrophy). It is an intriguing question how the two aspects of the disease, inflammation and neurodegeneration, influence each other. On the one hand, it is well known that the recruitment of peripheral immune cells can activate signaling cascades leading to neurodegeneration. In EAE, various aspects of neurodegeneration can be found, including synaptic degeneration, dendritic spine loss [64,65], alterations of synaptic plasticity [66], or the loss of lower motor neurons [67]. Siffrin nicely demonstrated that in EAE, the direct interaction of myelin oligodendrocyte glycoprotein (MOG)-specific Th17 and neuronal cells in demyelinating lesions is associated with extensive axonal damage [68]. Thus, the anti-inflammatory activity of sipinimod might results in less severe neurodegeneration and in consequence, amelioration of brain atrophy and disease progression. 

On the other hand, it has been shown that siponimod readily crosses the blood brain barrier and therefore potentially exerts beneficial effects by a direct interaction with brain cells. For example, findings from preclinical studies suggest that siponimod prevents synaptic neurodegeneration [69] and has the potential to promote remyelination in the CNS [70]. Additionally, siponimod was shown to modulate biological pathways involved in cell survival with subsequent attenuation of demyelination, in a mouse model [71]. It is therefore also possible that siponimod prevents brain atrophy and disease progression by directly interfering with brain cells, such as astrocytes, microglia, oligodendrocytes or neurons. Both aspects have recently been addressed in a commentary [72]. A third aspect, however, has not been discussed so far: The observed anti-inflammatory effects of siponimod in the EXPAND trial might be due to a primary CNS protective effect. 

In the following chapter we will list evidence that (i) degenerative events within the CNS can trigger the development of focal, inflammatory lesions and (ii) that siponimod can potentially ameliorate such degenerative processes and, in consequence, ameliorates the formation of new, focal inflammatory brain lesions. 

### 2.4. Degenerative CNS Events can Trigger Peripheral Immune Cell Recruitment

A central question of inflammatory lesion development in MS and its various animal models remains to be answered: What is at the root of peripheral immune cell recruitment? Here we list evidence that brain intrinsic degenerative events (i.e., degeneration of neurons, myelin destruction, endothelial dysfunction or reactive gliosis) might “guide” peripheral immune cells into the brain, and thus trigger the formation of new inflammatory lesions. The fact that neurodegenerative events can trigger the recruitment of peripheral immune cells into the CNS is well known and has been shown in experimental paradigms of peripheral facial nerve injury [73], cortical cryoinjury or eyeball enucleation [74]. 

Several reports suggest that in EAE, the archetypical model of autoimmune driven CNS inflammation, the recruitment of peripheral lymphocytes into the CNS, is a secondary and not a primary event. For example, the stress-associated transcription factor (ATF3) is induced in sensory brain stem neurons prior to T cell infiltration [75], suggesting that sensory neurons are, in fact, “activated” (manifesting with ATF3 induction) very soon after EAE immunization. Such a neuronal activation might be mediated by complete Freud adjuvans (CFA) and/or pertussis toxin (PTX), which are both administered with myelin peptides to induce encephalitogenic T cell development. CFA/PTX, besides its peripheral effects on immune cells [76], indeed can activate signaling cascades which are central for the recruitment of peripheral immune cells. For example, it has been shown that CFA/PTX increases CNS vascular permeability [77,78], at least in susceptible species [79], primes the choroid plexus as a gateway for leukocytes to enter the CNS [80,81], or induces chemokine expression of brain and spinal cord endothelial cells [82]. Although further work has to be done to understand in detail how EAE lesions develop, there is basic evidence suggesting that brain intrinsic degenerative events are involved in this process. Of note, in EAE microglia activation occurs early in the EAE brain and spinal cord, even before the appearance of severe motor deficits [83,84]. 

In classical EAE models (i.e., MOG_35–55_-induced (Myelin oligodendrocyte glycoprotein) EAE in C57Bl6 mice) inflammatory lesions predominantly develop within the spinal cord and cerebellum, whereas the forebrain is far less severely affected. Our group and others recently demonstrated that primary oligodendrocyte degeneration can trigger peripheral immune cell recruitment into the forebrain [85,86,87,88]. In the Scheld study, forebrain oligodendrocyte apoptosis was first induced by a 3-week intoxication with cuprizone, followed by the induction of encephalitogenic T-cell formation in peripheral lymphoid organs via the immunization with the myelin oligodendrocyte glycoprotein 35–55 peptide (MOG_35–55_ + CFA/PTX for the induction of active EAE). As already pointed out above, in MOG_35–55_-induced EAE, inflammatory infiltrates are found predominantly in the spinal cord and cerebellum, whereas the forebrain is largely spared. However, when we combined the cuprizone and the EAE model (called Cup/EAE model; see Figure 3), perivascular inflammatory infiltrates were found at several topographical sites including the corpus callosum, cortex and subcortical structures [85,88]. On the histopathological level, such infiltrates were characterized by the destruction of the perivascular glia limitans, monocyte, lymphocyte and granulocyte extravasation as well as focal axonal injury. Our findings and the work of others [86,87], therefore clearly illustrates the significance of brain-intrinsic degenerative cascades for immune cell recruitment and, possibly, MS lesion formation. 

### 2.5. Siponimod Ameliorates Degenerative Brain Events

As already pointed out above, siponimod readily crosses the blood brain barrier, and the receptors of siponimod, S1P1 and S1P5, are expressed by neural cells such as astrocytes [31], oligodendrocytes [32,33], microglia or neurons [89,90]. In order to assess whether siponimod has direct neuronal effects, in one elegant study, the drug was delivered directly into the brain by means of continuous intracerebroventricular infusion. While such a siponimod treatment strategy ameliorated the EAE disease score, it did not affect peripheral CD3^+^ cell counts [69]. Of note, astrocytosis and microgliosis as well as neuronal loss were less severe in siponimod-treated mice, and IL6 secretion was ameliorated in cultured microglia by siponimod. From these results it was concluded that siponimod is neuroprotective (i.e., it prevents the loss of neurons [69], probably by the modulation of microglia cell function). Furthermore, it has been shown using an organotypic slice culture model, that siponimod attenuates lysophosphatidylcholine-induced demyelination, suggesting that siponimod ameliorates as well oligodendroglia-degeneration. This assumption is in line with a recent report showing that siponimod decreases oligodendrocyte loss and demyelination in the cuprizone model [91]. 

## 3. Concluding Remarks

In MS patients, lesion development might start with a primary CNS degenerative event, which triggers the recruitment of peripheral immune cells into the CNS, resulting in the formation of focal inflammatory CNS lesions. Figure 4 summarizes our proposed mode of action for siponimod. First, siponimod modulates the activation status of microglia and/or astrocytes by either a direct modulation of glia reactivity or, alternatively, by ameliorating neuronal and/or oligodendrocyte injury. Second, the modulation of astrocytes and microglia results in a stabilization of the blood brain barrier and third, in consequence, leads to less severe peripheral immune cell recruitment. This concept does not exclude the possibility that additional peripheral immune cells are trapped within the peripheral lymphoid organs and, therefore, cannot be recruited into the CNS (fourth). The reduced inflammatory activity observed in the EXPAND study might, thus, be due to central protective effects of siponimod. A better understanding of such proposed degenerative processes would allow the development of drugs with both, anti-inflammatory and neuroprotective properties. For example, sex steroids such as 17β-estradiol have been shown to mediate both, neuroprotective and anti-inflammatory effects during EAE [92]. Additionally, endoplasmic reticulum stress responses have been shown to mediate both, toxin-induced oligodendrocyte degeneration [93], T cell differentiation [94] or apoptosis [95]. 

## Figures and Tables

**Figure 2 cells-08-00024-f002:**
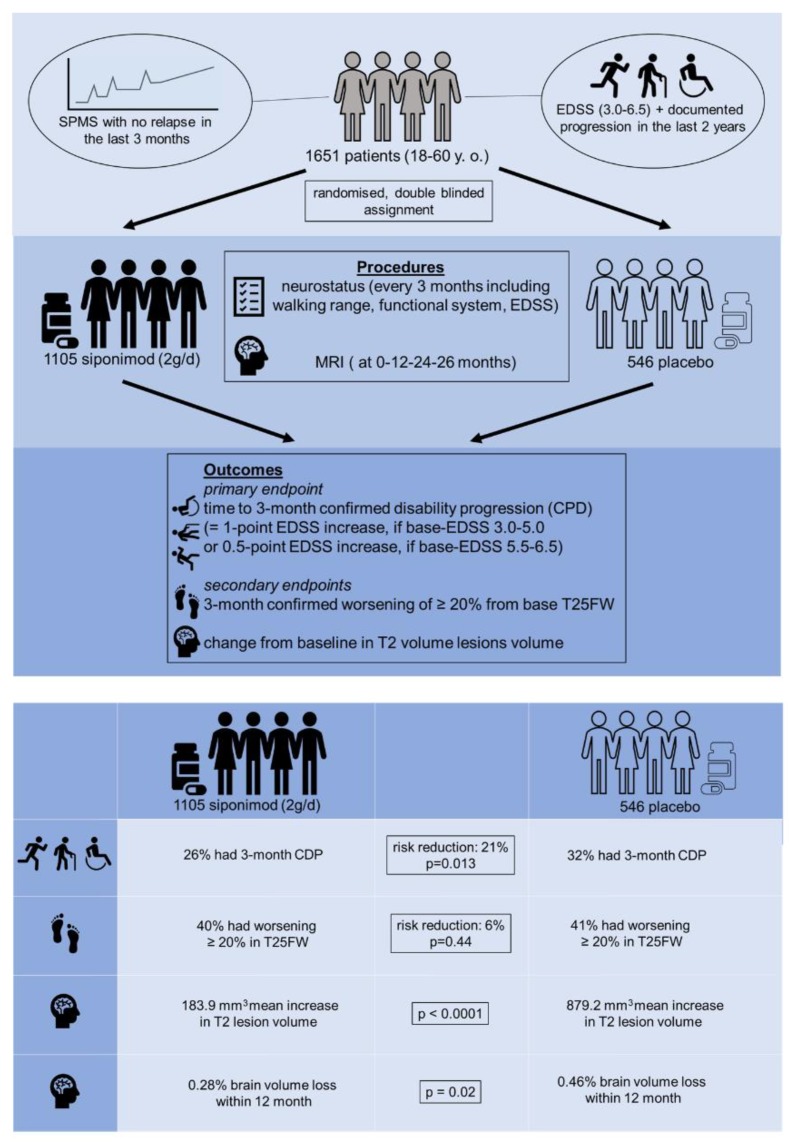
EXPAND-study-design: Schematic illustrating the principal design of the EXPAND trial. EDSS (Expanded Disability Status Scale); CDP (confirmed disability progression); T25W (timed 25-foot walk). Note that siponimod exerts anti-inflammatory and neuroprotective effects.

**Figure 3 cells-08-00024-f003:**
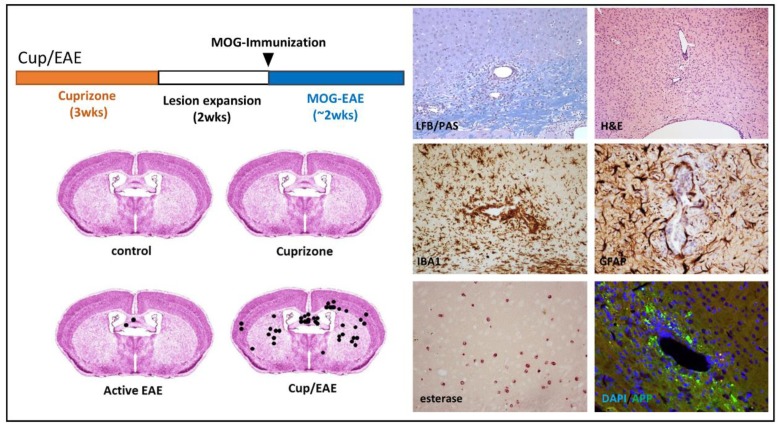
The Cup/EAE-model (Cuprizone/Experimental autoimmune encephalomyelitis): On the left upper part, the principal experimental setup of a classical Cup/EAE experiment is illustrated. During the first three weeks, animals were intoxicated with cuprizone (0.25%; orange bar), followed by two weeks on normal chow (white bar). At the beginning of week six (arrowhead) animals were immunized with MOG_35–55_ peptide + CFA/PTX (complete freund’s adjuvant/pertusistoxin). The lower images demonstrate the number and local distribution of perivascular infiltrations. On the right site, histopathological characteristics of such perivascular infiltrations are demonstrated. Luxol fast blue (LFB)/periodic acid-Schiff (PAS); Anti-ionized calcium-binding molecule 1 (IBA1); Anti-glial fibrillary acidic protein (GFAP); Anti-amyloid beta (A4) precursor protein (APP).

**Figure 4 cells-08-00024-f004:**
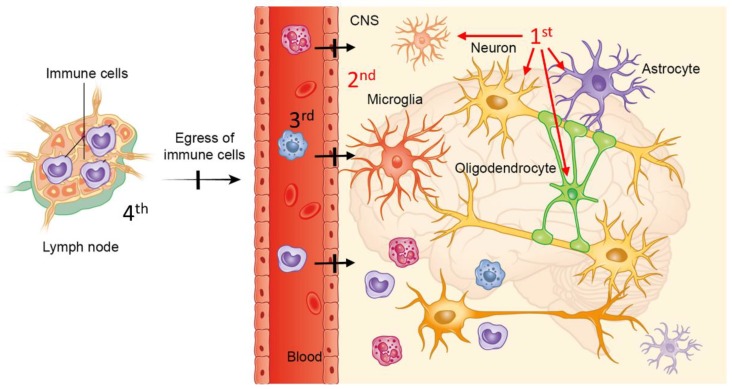
Proposed mode of action of siponimod: Schematic illustrating our proposed mode of action of siponimod. Note that following this concept, the anti-inflammatory activity of siponimod is at least in part due to direct interactions with brain cells. The copyright license of this illustration is with the authors.

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
