# Peer review of "Mechanism of Siponimod: Anti-Inflammatory and Neuroprotective Mode of Action"

_cells, 2019, doi:10.3390/cells8010024_

Round 1

Reviewer 1 Report

Work by Behrangi et al. presents a review that revolves around mechanisms of siponimod treatment.  It proposes several potential mechanisms of action and discusses applicability in the context of MS. As a secondary theme, authors develop an idea that MS may originate within the CNS suggesting that recruitment of peripheral leukocytes is a consequence rather than the cause of MS. The origin of the MS is constantly discussed and there a few well-developed hypotheses. Although without naming it, authors favor the inside-out hypothesis. The paper is timely and will be of interest to the broad audience of MS professionals at many levels.

Weaknesses

The title is misleading. In my opinion, it should include the name of the treatment that is being discussed.

It seems that at least two different people wrote the paper. The abstract and introduction use convoluted, difficult to understand language while the rest of the paper is clearly written and easy to follow.

I would suggest using short sentences with clear messages. The idea of neurodegeneration as a trigger of MS is not novel. Therefore, the discussion of outside-in and inside-out models of MS, in my opinion, will help to organize this review.

Author Response

#1 Reviewer

Work by Behrangi et al. presents a review that revolves around mechanisms of siponimod treatment.  It proposes several potential mechanisms of action and discusses applicability in the context of MS. As a secondary theme, authors develop an idea that MS may originate within the CNS suggesting that recruitment of peripheral leukocytes is a consequence rather than the cause of MS. The origin of the MS is constantly discussed and there a few well-developed hypotheses. Although without naming it, authors favor the inside-out hypothesis. The paper is timely and will be of interest to the broad audience of MS professionals at many levels.

Q1: The title is misleading. In my opinion, it should include the name of the treatment that is being discussed.

A1: Thank you for the comment. We have adopted the title in the revised version of the manuscript

Q2: It seems that at least two different people wrote the paper. The abstract and introduction use convoluted, difficult to understand language while the rest of the paper is clearly written and easy to follow.

A2: Thank you for the comment. We have adopted the abstract and introduction and hope that both parts are now easier to understand

Q3: I would suggest using short sentences with clear messages. The idea of neurodegeneration as a trigger of MS is not novel. Therefore, the discussion of outside-in and inside-out models of MS, in my opinion, will help to organize this review.

A3: We deliberately ignore the term “outside-in and inside-out”. Most people argue that after brain damage, CNS antigens gain access to regional lymph node (for example to the superficial and deep cervical lymph nodes) which then results in an encephalitogenic autoimmune response (whatever the antigen might be). However, this is NOT what we suggest. Actually, this article does not suggest any mechanism how autoreactive B or T cells develop. We rather argue that once autoreactive B or T cells have developed in the periphery, focal degenerative processes within the CNS trigger their recruitment into the CNS and, thus, the inflammatory lesion development. We have tried to point this out at the very end of the manuscript by stating “Of note, we do not propose that brain degenerative events trigger the development of encephalitogenic immune cells but rather suggest that once encephalitogenic immune cells are present in the blood in relevant concentrations, CNS-intrinsic events might trigger their central recruitment and, thus, the development of focal inflammatory lesions.” We have moved this statement at the very beginning of the manuscript to avoid any confusions.

Reviewer 2 Report

This is an interesting review that proposes, “neurodegeneration can trigger the formation of new, focal inflammatory lesions” in MS, and uses data based on the mechanism of action, receptor selectivity, and clinical trial data with siponimod to support this assumption.  The authors argue that this may offer a unique perspective on how we view disease progression in MS and may lead to alternative therapeutic strategies.

As a whole, the MS is of interest.  However, the authors could strengthen argument by including a section in the review proposed ways to model the impact that “degenerative intervention” may have on subsequent inflammatory and demyelinating events.  Further, the authors, in the introduction, propose that embracing the perspective that “degeneration” may precede inflammation-induced demyelination may lead to alternative therapeutic strategies to treat people suffering from MS.  However, the authors do not include in their review how this could be specifically addressed.  Therefore, considering the implications of the idea it would be of interest to expand the review to include a discussion on these two topics.

Minor comments:

Line 52: It would be of interest to temper the hyperbole of this exclamatory state into one that explains how this theory may have “direct therapeutic consequences”.

Lines 59-62:  As written this is a very awkward paragraph which should be further developed to be more clear.

Figure 1: In the figure, it should be made clear that the numbers following the colon describe references.  For example, S1P1: 21-23 could be S1P1: Ref 21-23, (refereeing to figure legend) or S1P1: ref [86,87, and 32] (refereeing to the actual reference list).’

Lines 199-213: It is not clear why this part of the text is “Center Justified”

Line 213: There appears to be a good amount of lost text that make this part of the manuscript difficult to impossible to interpret.

Line 240-251: It is not clear why this part of the text is “Center Justified”.  Further, it would be of interest to expand on the statement, “A better understanding of such proposed degenerative processes would allow the development…”

Author Response

#2 Reviewer

Q1: This is an interesting review that proposes, “neurodegeneration can trigger the formation of new, focal inflammatory lesions” in MS, and uses data based on the mechanism of action, receptor selectivity, and clinical trial data with siponimod to support this assumption.  The authors argue that this may offer a unique perspective on how we view disease progression in MS and may lead to alternative therapeutic strategies. As a whole, the MS is of interest.  However, the authors could strengthen argument by including a section in the review proposed ways to model the impact that “degenerative intervention” may have on subsequent inflammatory and demyelinating events.  Further, the authors, in the introduction, propose that embracing the perspective that “degeneration” may precede inflammation-induced demyelination may lead to alternative therapeutic strategies to treat people suffering from MS.  However, the authors do not include in their review how this could be specifically addressed.  Therefore, considering the implications of the idea it would be of interest to expand the review to include a discussion on these two topics.

A1: Thank you for this fruitful comment. We have adopted the manuscript accordingly, highlighting the combined anti-inflammatory and neuroprotective role of estrogens and endoplasmic reticulum stress responses.

Q2: Line 52: It would be of interest to temper the hyperbole of this exclamatory state into one that explains how this theory may have “direct therapeutic consequences”.
A2: Thank you for this comment. We have tried or best to point this out more clearly.

Q3: Lines 59-62:  As written this is a very awkward paragraph which should be further developed to be more clear.

A3: The section has been adopted accordingly.

Q4: Figure 1: In the figure, it should be made clear that the numbers following the colon describe references.  For example, S1P1: 21-23 could be S1P1: Ref 21-23, (refereeing to figure legend) or S1P1: ref [86,87, and 32] (refereeing to the actual reference list).’

A4: Thank you for this comment. The section has been adopted accordingly.

Q5: Lines 199-213: It is not clear why this part of the text is “Center Justified”

A5: Thank you for this comment. This was a mistake

Q6: Line 213: There appears to be a good amount of lost text that make this part of the manuscript difficult to impossible to interpret.
A7: We have carefully checked this part of the manuscript. 

Q7: Line 240-251: It is not clear why this part of the text is “Center Justified”.  Further, it would be of interest to expand on the statement, “A better understanding of such proposed degenerative processes would allow the development…”

A7: Thank you for this comment. The section has been adopted accordingly.